# Interleaved Electroactive Molecules into LDH Working on Both Electrodes of an Aqueous Battery-Type Device

**DOI:** 10.3390/molecules28031006

**Published:** 2023-01-19

**Authors:** Julien Sarmet, Fabrice Leroux, Christine Taviot-Gueho, Patrick Gerlach, Camille Douard, Thierry Brousse, Gwenaëlle Toussaint, Philippe Stevens

**Affiliations:** 1Université Clermont Auvergne, Clermont Auvergne INP, CNRS, Institut Pascal, F-63000 Clermont-Ferrand, France; 2Nantes Université, CNRS, Institut des Matériaux de Nantes Jean Rouxel, IMN, 2 rue de la Houssinière BP32229, CEDEX 3, F-44322 Nantes, France; 3Réseau sur le Stockage Electrochimique de l’Energie (RS2E), CNRS FR 3459, 33 rue Saint Leu, CEDEX, F-80039 Amiens, France; 4EDF R&D, Department LME, Avenue des Renardières, CEDEX, F-77818 Moret-sur-Loing, France

**Keywords:** layered double hydroxide, riboflavin, LDH-riboflavin, LDH-ferrocene battery-type device

## Abstract

By selecting two electroactive species immobilized in a layered double hydroxide backbone (LDH) host, one able to act as a positive electrode material and the other as a negative one, it was possible to match their capacity to design an innovative energy storage device. Each electrode material is based on electroactive species, riboflavin phosphate (RF) on one side and ferrocene carboxylate (FCm) on the other, both interleaved into a layered double hydroxide (LDH) host structure to avoid any possible molecule migration and instability. The intercalation of the electroactive guest molecules is demonstrated by X-ray diffraction with the observation of an interlayer LDH spacing of about 2 nm in each case. When successfully hosted into LDH interlayer space, the electrochemical behavior of each hybrid assembly was scrutinized separately in aqueous electrolyte to characterize the redox reaction occurring upon cycling and found to be a rapid faradic type. Both electrode materials were placed face to face to achieve a new aqueous battery (16C rate) that provides a first cycle-capacity of about 7 mAh per gram of working electrode material LDH/FCm at 10 mV/s over a voltage window of 2.2 V in 1M sodium acetate, thus validating the hybrid LDH host approach on both electrode materials even if the cyclability of the assembly has not yet been met.

## 1. Introduction

After years of research devoted to electrochemical storage systems and to prepare the post-lithium technology, many efforts have been paid on the development of sustainable and renewable-energy sources which are be cost-efficient and safe [1,2]. Indeed, there is a great need to find new electrochemical, versatile and robust storage systems and, more specifically, to regulate the fluctuations in the production and consumption of equipment connected to electrical networks. As mentioned, such systems must rely on materials with fast or even very fast charge transfer, and all the different parts composing storage systems are scrutinized for optimization [3]. Recent trends focus on rechargeable batteries using aqueous electrolyte to respond for instance to demands for stationary applications [2], developing new batteries based on Na^+^ cation dissociation and diffusion in aqueous electrolyte [4], Mg^2+^ [5] but also Al^3+^ [6].

Among the categories of materials that are able to meet these specifications, there are electrochemically active inorganic frameworks and, as far as metal hydroxides are concerned, most of the recent studies are on cobalt–nickel double hydroxides for zinc-ion batteries (ZIB) in aqueous electrolyte [7,8]. Some alternatives use metal hydroxides as an artificial solid electrolyte interface in ZIB as hydrotalcite [9] or NiCo [10] to avoid zinc dendrites, while others are combined in asymmetric devices [11,12], and some relative oxides are also developed for supercapacitors (SC), such as aminophenol-modified zinc oxide [13] or substituted hydroxide Ce_x_Ni_1-x_O_2_ [14]. A little change of paradigm compared to what is today studied in metal hydroxide concerning the cation composition and substitution is here to focus on their ability to host electroactive organic species as pioneered on anthraquinone-sulfonate (AQS) for sensors [15] but not as a possible energy-source so far. In fact, organic molecules are also studied in electrochemical storage systems as redox flow batteries (RFB) or supercapacitors (SC) [16]. Most electroactive molecules studied today are derived from metals, which are often expensive and even toxic [17]. However, among the most promising molecules are those derived from quinone [18,19], nitroxide radicals [20,21], flavins [22,23] or ferrocene [24,25,26], while some of them are sustainable, exemplified by biomass-derived energy-storage systems such as alizarin [1] or alloxazine [4]. In this work we focus on the electroactivity of flavin derivatives, which have been widely studied in the biological field as a sensor [27], where they participate in catalyzing a large number of redox reaction [28]. For example, riboflavin (vitamin B2) acts as an electroactive precursor for the biochemical synthesis of nucleotides. However, only a few studies have shown the potential of these molecules for electrochemical energy storage devices [29], adsorbed on the surface of materials such as graphite or singled-wall carbon nanotubes (SWCNT), or for lithium-ion batteries [30]. In addition, they have been used to enhance the oxygen evolution reaction (OER) combined with Ni-MOF [31] or in an organic flow desalination battery [32].

The molecular backbone of riboflavin (RF) presents a conjugated iso-alloxazine ring which, by the donation or withdrawal of electrons, gives rise to various states, including oxidized, reduced or radical semi-quinone. Therefore, the electrochemical process takes place via a two-electron process or through the semi-quinone radical (Figure 1) in separate one-electron processes [33,34]. Overall, the associated mid-point redox potential localized at around −0.5 V vs. SCE makes this organic molecule of interest as a negative electrode.

Comparatively, ferrocene type molecules, once adsorbed on carbon nanotubes [35] mixed with reduced graphene oxide (rGO) [36] or ferrocene-based coordination polymer [37], present a redox activity with a potential of +0.5 V vs. SCE, thus making them interesting as positive electrodes for an asymmetric design involving two redox active molecules operated in complementary potential windows. These kinds of molecules have been used as an electrochemical sensor of biomolecules [38] or metals [39]. Other articles demonstrate the possibility of increasing the capacity of electricity storage using 3D graphene-oxide/ferrocene redox-active composites for supercapacitor application [40].

The combination of the two materials to form an electrochemical system in an aqueous medium is tempting, but they need to be trapped or contained in order to avoid the migration/dissolution of the electroactive molecules. Other than surface adsorption, the incorporation of electroactive molecules into a host structure is a means to stabilize, in the first cycles, the electrode, but it is difficult to predict their response over time. Layered double hydroxide (LDH) is a class of materials known to easily accommodate anions of all kinds for a wide variety of applications [41]. They consist of edge-sharing metal hydroxide octahedra of brucite-type structure in which a partial substitution of divalent to trivalent cations produces an excess of a positive charge of layers that is compensated by the presence of anions between the layers. Large cumbersome species have been hosted into LDH interlayer spaces to provide stability and prevent their migration when exposed to external stresses or dispersed in a polymer for red-emitting quantum dots, for example [42]. In the same vein, we focus here with a common LDH, the hydrotalcite of general composition Mg_2_Al(OH)_6_-A·nH_2_O hereafter labelled as Mg_2_Al-A, where A would be the electroactive interleaved molecule. Ferrocene (FC) di- and mono-sulfonate and di and mono-carboxylate molecules have in the past been hosted between LDH layers, and the whole hybrid assembly was studied for glucose detection [43]. Other publications report the intercalation of ferrocenesulfonate or ferrocenecarboxylate anions by coprecipitation in Zn-Cr or Zn-Al LDH [44,45,46]. Moreover, neutral ferrocene and ferrocene methanol were immobilized, respectively, by inclusion within cyclodextrin cavities grafted into LDH [47] or by simple adsorption on the LDH particles [48]. In addition, hybrid LDH containing organic electroactive molecules, such as anthraquinone mono and disulfonate [49], 2,2′-azinobis 3-ethylbenzothiazoline-6-sulfonate (ABTS) [50] and nitroxide [49], have been studied but no flavin-type molecules as of yet. The presence of the phosphonate group at the end of the chain of the riboflavin molecule makes its incorporation into an LDH phase possible. The association of two LDH matrices intercalated by electroactive molecules cycling in an aqueous media has never been reported in the literature and seems to be an interesting perspective in line with the current green chemistry principles.

The two series of hybrid materials, RF/LDH and FC/LDH, have been characterized by X-ray diffraction and their electrochemical behavior evaluated by cyclic voltammetry. In the following hybrid materials, Mg_2_Al LDH interleaved with riboflavin-5′- phosphate, ferrocene mono- (FCm) or di-carboxylate (FCd) are named Mg_2_Al-RF, Mg_2_Al-FCm and Mg_2_Al-FCd, respectively. Both electrode materials are form an innovative aqueous battery: Mg_2_Al-RF//Mg_2_Al-FCm or –FCd.

## 2. Results

### 2.1. Intercalation of Electroactive Molecules in LDH Matrix

The XRD patterns of as-prepared hybrid samples present diffraction peaks which can be interpreted by a series of a large number of (00l) diffraction lines and few (hk0) diffraction peaks, which themselves can be indexed in the rhombohedral space group R3m, typical for LDH-based systems (Figure 2). The comparison of the intensity of the peaks between (00l) and (hk0) suggest that the diffraction feature is dominated by the stacking along the perpendicular axis to the layers. The value of the basal spacing is calculated by applying the Bragg relationship to the position of the diffraction peaks (00l). From a basal spacing of about d = 0.8 nm present in Mg_2_Al-Cl, the value is larger for the hybrid materials for accommodating the guest molecules, but their crystallinity decreases noticeably. The interlayer distance calculated for Mg_2_Al-FCm and LDH-FCd is in accordance with the literature [49], with monolayer arrangement for FCd having two negative charges tethered to the hydroxyl LDH layers, while a double layer or, most probably, some kind of interpenetrated herring-bone arrangement for FCm. For the cumbersome guest RF whose embedding into LDH has never been reported so far, the basal spacing of the new resulting hybrid material is of 2.3 nm, corresponding to a monolayer arrangement. The XRD of the LDH-RF phase shows low crystallinity in comparison with LDH-FC phases, which is due to the size of the molecule making it difficult to be accommodated.

CHNS (Table 1) and TGA (Appendix A) analyses completed by infrared spectra (Appendix A) give the exact composition of the synthesized materials, which are as follows: LDH-RF: Mg_2_Al(OH)_6_(CO_3_^2−^)_0.185_(RF)_0.630_·3.1H_2_O; LDH-FCm: Mg_2_Al(OH)_6_(FCm)_1.0_·2.8H_2_O; and LDH-FCd: Mg_2_Al(OH)_6_(FCd)_0.5_·2.9H_2_O. The intercalation of RF in LDH phase is not complete, according to elemental analyses, and the total charge balance is ensured by the co-intercalation of carbonate anions, as evidenced by the mass spectrum (peak at 44 g/mol in Appendix A).

The scanning electron microscopy image (Appendix A) reveals, for both hybrid phases, a similar shape in aggregates with a large distribution of size made of agglomerated and rather ill-defined platelets.

### 2.2. Electrochemical Behavior of LDH-RF

By using the same proportion of the electroactive molecule RF in a composite electrode made of a mixture of carbon black and PTFE (see experimental section) with and without the molecules incorporated into the LDH host, it is possible to estimate the effect of such confinement on the electrochemical behavior.

The cyclic voltammograms of the molecule RF on its own display on the first cycle a pronounced reduction peak at −0.49 V vs. Ag/AgCl and an anodic re-oxidation around −0.25 V (Figure 3a). Upon further cycling, these two peaks are progressively shifted to more negative values and are stabilized at −0.65 V and −0.35 V vs. Ag/AgCl, respectively, after a few hundred cycles. The reversible redox waves separate upon cycling, ΔE_1_ = 0.24 vs. ΔE_100_ = 0.30 V, underlining that the redox process is increasingly more difficult; this is due to the fact that part of the riboflavin molecules are no longer involved in reversible redox processes, thus causing a large decrease in capacity.

When hosted in an LDH, and in comparison to the free RF, the cyclic voltammograms of LDH-RF show a reversible redox wave shifted to more negative values, −0.7 V and −0.4 V vs. Ag/AgCl during the first cycle, in reduction and re-oxidation, respectively. Such shift of cathodic and anodic peaks when the electroactive molecules are hosted in a material is often observed, for example in zirconium phosphate [34], and results from the interactions between the electroactive molecules and the host matrix.

The theoretical capacity of LDH-RF is 71.6 mAh/g based on 2e^−^/formula weight and M = 533.4 g/mol. In the first cycle, the capacity is 7 mAh/g (Appendix A) for RF and 8 mAh/g (Figure 3d) for LDH-RF only, which corresponds to 9.7% and 11.2% theoretical capacity, respectively. During the cycling, the capacity decreases rapidly down to 3.7 mAh/g for the RF sample with a poor columbic efficiency of around 40%, whilst the capacity for LDH-RF increases up to 16 mAh/g (22.3% theoretical capacity) with a coulombic efficiency of 90% after the 500th cycle.

The increase in capacity upon cycling can be explained by an interface between electrolyte and LDH material becoming more active. Such an increase in redox activity is most probably due to an easier access to the sites by an increase in the wettability of the composite material, thus inducing an easier ingress of the electrolyte salt in the interface vicinity. A de-agglomeration of particles, exposing more sites and creating more interface, could also possibly explain the evolution of capacity, but as this would cause a loss of electrical contact, this hypothesis is not considered. However, the potential separation between cathodic and anodic peak increases during cycling to reach a delta of 0.45 V after the 500th cycle, making the redox process less reversible, mostly due to a shift in the reduction step. In a full cell, this would result in a decrease in discharge voltage and therefore a decrease in both voltaic and energy efficiency.

By decreasing the scan rate down to 1 mV/s, such potential separation decreases significantly underlining that the redox reaction is kinetically limited. The redox reduction wave is still centered at −0.5 V vs. Ag/AgCl after 25 cycles (Figure 3c). The associated capacity value obtained in the first cycle is of 30.4 mAh/g (42.5 % theoretical capacity) and 72.8 mAh/g (101% theoretical capacity) after the 500th cycle (Appendix A). This last value, close to the theoretical capacity, is a direct consequence of water electrolyte reactions which occur below −0.9 V vs. Ag/AgCl. By combining these results to plot the capacity evolution as a function of v ^−0.5^ (where v is the scan rate in mV.s^−1^, Appendix A), it is possible to characterize the electrochemical process knowing that the maximum surface capacitive response is extrapolated when the scan rate tends to infinite, i.e., v ^−0.5^ → 0. This surface capacitive response is zero here, underlining that the electrochemical process is not capacitive at all (Q_surface_ = 0) but is faradic in nature, as expected by its strong dependence with respect to the scan rate.

Different electrolyte salts have been tested (Table 2, Appendix A), and the best capacities obtained at the first cycle are the consequences of the electrolyte anion size and the diffusion of electrolyte anion in the interlayer space (${\mathrm{NO}}_{3}^{-}$ < ${\mathrm{SO}}_{4}^{2-}$ < ${\mathrm{ClO}}_{4}^{-}$ < ${\mathrm{CH}}_{3}{\mathrm{COO}}^{-})$. Regardless of the nature of the salt, all the plots of Q = f(v ^−0.5^) present a surface capacity close to 0 mAh/g (Appendix A).

The best capacities are obtained with sodium nitrate, but they are associated with a poor coulombic efficiency. This could imply that the measured capacity is perhaps not only attributed to LDH-RF but associated with a parasitic reaction that would also provide capacity in reduction (e.g., reaction involving H_2_O/H_2_ and ${\mathrm{NO}}_{3}^{-}/{\mathrm{NO}}_{2}^{-})$. For the cycling tests at 10 mV/s, capacities increase for sodium acetate, sodium perchlorate and lithium sulfate and give the better efficiency around 90% after 500th cycle.

Moreover, the layers composed by Mg^2+^ and Al^3+^ cations remain stable since they are electrochemically inactive in the operating conditions (potential range and electrolyte). Consequently, the redox process generated in the interlayer space destabilizes the LDH-organic molecule assembly, leading to the de-intercalation of the organic molecules and to the ingress of the electrolyte anions to equilibrate the total charge of the hybrid framework. Post-mortem XRD analyses (Figure 4 and Appendix A) show that the electroactive anions have left the interlayer space whilst the MgAl-LDH matrix is partly preserved upon cycling. On the first cycle, LDH-FCm is stable, and the molecule remains in the interlayer space. After the 5th cycle, most of the electroactive anions have moved out of the interlayer space and have been replaced by the electrolyte salt anion. In addition, SEM images of LDH-RF and LDH-FCm (Appendix A) show the conservation of the electrode structure. By combining these results with EDX mapping (Appendix A) it is possible to observe FCm molecules coming out of the LDH structure. Indeed, the iron element on the pristine paste is located at the same places as those for Mg and Al in the LDH structure. During the cycling and, in particular, after the 50th cycle, there is no more correlation between iron element and the two others (Mg, Al), even if the structure of the LDH phase is preserved (concomitant presence of Mg and Al and evidenced by XRD analyses). Moreover, the results collected at 1 mV/s and reported in Appendix A underline the possibility of reaching 80% theoretical capacity with sodium nitrate. Sodium acetate and lithium perchlorate provide both the best capacity values during cycling with 73.3 and 97.8 mAh/g after the 25th cycle, respectively.

### 2.3. Electrochemical Behavior of LDH-FCm and LDH-FCd

Theoretical capacities for LDH-FCm and LDH-FCd are 58.6 and 26.7 mAh/g, respectively, by assuming a transfer of 1 and 0.5 e^−^ per formula weight determined before by CHNS analysis. Different salts are tested for the two hybrid materials, sodium acetate, sodium and lithium perchlorate, sodium and lithium sulfate (Figure 5). LDH-FCd samples provide a poor capacity during cycling at 10 mV/s (Table 2 and Figure 5a) regardless of the nature of the salt. Figure 5a shows no redox reaction for sodium acetate and the two sulfate-based electrolytes whilst a low current density is observed for the two perchlorate electrolytes. This absence of response seems to be correlated with the impossibility of accessing the redox centers inside the hybrid host. The most probable explanation is that FCd, attached by its two carboxylate functions and playing the role of pillar, prevents the access of the electrolyte anion into the interlayer space. Indeed, by reducing the scan rate down to 1 mV/s (Appendix A), the capacity increases up to 53% theoretical capacity for sodium sulfate, which underlines the kinetic limitation of the redox reaction for the interleaved guest molecules.

In contrast with this result, the LDH-FCm system presents a good capacity in the first cycle at 10 mV/s with 33%, 69% and up to 98% theoretical capacity for sodium acetate, lithium sulfate and lithium perchlorate, respectively (Table 3 and Figure 5b). It shows that the diffusion of the electrolyte is better than for the hybrid LDH-FCd. Using lithium sulfate salt provides a pronounced reversible redox peak. By decreasing the scan rate down to 1 mV/s (Appendix A) the theoretical capacity is reached in the first cycle for all the electrolytes, but it is not stable upon cycling and it fades rapidly.

### 2.4. Test in Aqueous Battery

A full LDH-FCm//LDH-RF cell was assembled using LDH-RF at the negative electrode and LDH-FCm at the positive electrode. Care was taken to equilibrate the charge in both electrodes, i.e., a loading in mAh at the negative electrode equivalent to that of the positive electrode and, therefore, an optimization of the quantities of electrode materials. To scrutinize the electrochemical performance, the redox reactions are recorded using different electrolyte salts using linear cyclic voltammetry at 10 mV/s (equivalent to a 16–18C rate of charge or discharge). Figure 6a compares the first cycle of the two materials taken separately and that of their assembly within the same device. Figure 6a clearly indicates the possibility of combining both of them in an aqueous battery design.

Using different electrolyte salts, the potential of the working electrode LDH-FCm is imposed by the electrochemical test using CV mode, paying special attention to possible overlap (Appendix A). Indeed, a first oxidation at the positive electrode is imposed between the open circuit potential up to 1.0 V vs. Ag/AgCl and, simultaneously, the potential at the negative counter-electrode goes down to −1.5 V vs. Ag/AgCl, resulting in the acetate salt in a complete cell voltage ΔE = 1.0 − (−1.5) = 2.5 V on charge. However, when the reaction is reverse to the reduction of the positive electrode, a cell polarity inversion occurs when the WE is below 0.2 V vs. Ag/AgCl, since the potential of the negative electrode rises steeply above −0.5 V vs. Ag/AgCl, which is deleterious for the complete cell. To avoid such deterioration, the CV return scan (discharge) is prevented from reaching 0.2 V vs. Ag/AgCl in a new experiment so that the potential sweeps of the WE occur between 1.0 V down to 0.2 V vs. Ag/AgCl (Appendix A).

In order to obtain a better understanding of the separate mechanisms involved at each electrode material, it is interesting to follow their potential evolution operating during the cycling of the full device (Figure 6b,c). In this set-up, a voltage for the complete cell of 2.2 V and 1.9 for sodium acetate and lithium perchlorate, respectively, are obtained during charge (Figure 6d).

The discharge capacity obtained on the first discharge is around 7 mAh/g in sodium acetate and (Figure 6e) and less than 2 mAh/g in lithium perchlorate (Figure 6f). In addition, different scan rates have been tested for this system. A charge that is close to 20–25 mAh/g (Figure 6g) during charge is obtained and 2 mV/s on the first discharge for the two aqueous electrolytes, but they rapidly fade upon cycling.

## 3. Materials and Methods

### 3.1. Reagents and Chemicals

For LDH-organic molecule synthesis, aluminum chloride hexahydrate (AlCl_3_·6H_2_O, 99%, Merck, Saint-Quentin-Fallavier, France) and magnesium chloride (MgCl_2_·6H_2_O, 98%, Merck, Saint-Quentin-Fallavier, France), sodium Hydroxide (NaOH, 98%, Acros, Geel, Belgium), ribloflavin-5′-phosphate (C_17_H_20_N_4_NaO_9_P, 98%, Merck, Saint-Quentin-Fallavier, France), ferrocene-carboxylic acid (C_11_H_10_FeO_2_, 97%, Merck, Saint-Quentin-Fallavier, France) and ferrocene-dicarboxylic acid (C_12_H_10_FeO_4_, 96%, Merck, Saint-Quentin-Fallavier, France) were used.

For electrode preparation, carbon black (SG, >99%, Superior Graphite Co., Sundsvall, Sweden) and polytetrafluoroethylene (PTFE, 60 wt% in water, Merck, Saint-Quentin-Fallavier, France) were used.

For electrochemical experiments sodium acetate (CH_3_COONa, ≥99%, Merck, Saint-Quentin-Fallavier, France), sodium nitrate (NaNO_3_, 99%, Merck, Saint-Quentin-Fallavier, France), sodium perchlorate (NaClO_4_, 99%, Merck), lithium perchlorate (LiClO_4_, 99%, Merck, Saint-Quentin-Fallavier, France), sodium sulfate (Na_2_SO_4_, 98%, Merck, Saint-Quentin-Fallavier, France) and lithium sulfate (Li_2_SO_4_, 99%, Merck, Saint-Quentin-Fallavier, France) were used. Deionized water was employed for all experiments.

### 3.2. Analysis

X-ray diffraction analyses were performed using a theta−theta PANalytical X’Pert Pro diffractometer equipped with a Cu anticathode (λKα1 = 1.540598 Å, λKα2 = 1.544426 Å) and an X’Celeretor detector. For the phase identification and refinement of the unit cell parameters of the series of samples, the X-ray patterns were recorded in the Bragg–Brentano geometry in the range of 2–80°/2θ with a step size of 0.325°/min three times.

Scanning electron microscopy (SEM) images and energy-dispersive X-ray spectroscopy (EDX) were recorded using a JSM-7500F field-emission scanning electron microscope operating at an acceleration voltage of 3 kV and at magnifications of ×1 K, ×20 K and ×50 K. Samples to be imaged were mounted on conductive carbon adhesive tabs and coated with a gold thin layer.

Fourier transform infrared (FT-IR) spectra were recorded in transmission mode using the KBr pellet technique (2% weight) with a Nicolet 5700 spectrometer from Thermo Scientific over the wavenumber domain of 400 to 4000 cm^−1^ with a scan number of 128 at a resolution of 4 cm^−1^.

Thermogravimetric analyses (TGA) and mass spectroscopy (MS) were performed using a SETSYS Evolution de Setaram 92 coupled with a Balzers mass analyzer under air flow 20 mL.min^−1^ in the temperature range of 25−1000 °C with a linear temperature ramp of 5 °C min^−1^.

Cyclic voltammetry (CV) curves were performed using a Biologic VMP3 with a three electrodes setup with LDH/SG/PTFE as a working electrode, Ag/AgCl (KCl sat) as a reference and platinum as a counter electrode. The aqueous electrolyte salt was sodium acetate, sodium perchlorate, sodium nitrate, sodium sulfate lithium perchlorate and lithium sulfate in a potential range between 0 to 0.9 V vs. Ag/AgCl for LDH-FC and −1 to 0 V for LDH-FCm. Complete cells were performed using LDH-FCm as a working electrode by imposing a potential range between 0.2 and 1.0 V vs. Ag/AgCl facing LDH-RF electrode material. Ag/AgCl (KCl sat) was placed between in order to follow the potential evolution of the two electrodes.

### 3.3. Material Synthesis

A layered double hydroxide of the type MgAl was prepared using a coprecipitation method similar to that described by Miyata. A total of 100 mL of deionized and decarbonated water was used and the preparation was carried out in a 250 mL reactor under a nitrogen atmosphere. Organic molecules (OM) were dissolved in this media and the pH was adjusted to 9.5 with 0.1 NaOH solution, following the ratio n_OM_ = 4n_Al_. A total of 25 mL of the aqueous metal chlorides solution (0.333 mol/L in MgCl_2_ and 0.167 mol/L in AlCl_3_) was added dropwise to the reactor at 25 °C and under magnetic stirring. The pH of the reaction mixture was kept constant at 9.5 by adding a 1 mol/L NaOH solution. After the total addition of the metal salts, the reaction mixture was aged at room temperature for 24 h. The precipitate was then washed three times with water and centrifuged. The solid was dried in oven during 24 h at 40 °C.

### 3.4. Electrode Preparation

In order to evaluate the electrochemical performance of the composite, the powder was shaped using a typical self-supporting electrode process. To do so, the active material, an electronically conductive additive (carbon black, Superior Graphite Co., >99%) and a polymer binder (PTFE, 10% wt. in solution in water) were mixed in 15 mL of ethanol so as to obtain 60, 30 and 10% mass of each component, respectively. The total mass loading of the electrode composite material was between 5 to 10 mg/cm^2^. The mixture was then stirred and heated to 60–70 °C to partially evaporate the solvent and obtain a homogeneous paste. The paste was then cold rolled into a sheet of approximately 150 µm thickness, before being dried in an oven at 60 °C overnight. Electrodes with a diameter of 10 mm were then cut from the paste and pressed into a stainless steel grid current collector (316 L, 0.160 mm, Saulas) for 1 min at 5 tons.

## 4. Conclusions

In summary, the intercalation of a novel vitamin B-derived molecule (riboflavin phosphate, RF) within a MgAl-LDH phase was achieved. Its intercalation was proven by means of XRD, IR, CHNS and TGA-MS analysis and demonstrates an increase of interlayer space to 2 nm on each case. Moreover, LDH-organic molecule compounds have been tested separately and provided a capacity around 9.0 mAh/g for LDH-RF and 19 mAh/g for LDH-Fcm in a 1 M sodium acetate solution at 10 mV/s. Capacity values and their evolutions depend on electrolyte salt used. No changes of LDH structure were observed during cycling even though organic molecules intercalated were expulsed out of the interlayer space and replaced by electrolyte anions. In addition, the electrochemical potential for its use as a negative electrode in an aqueous battery system which assembles two LDH material-based electrodes (LDH-RF//LDH-FCm) was tested by cyclic voltammetry. A charge capacity of 20–25 mAh/g was reached at 2 mV/s in sodium acetate and lithium perchlorate. At a faster cycling rate of 10 mV/s, the capacity decreases down to 6–10 mAh/g over a potential range of 2.2 V and 1.9 V, respectively. Despite the rapid fading in capacity after the first few cycles due to the exchange between the electroactive interleaved species and the anions from the electrolyte salt, this new organic–inorganic system operating in an aqueous electrolyte seems promising and future works should focus on the retention of RF within the LDH interlayer space to further advance this proof of concept.

## Figures and Tables

**Figure 1 molecules-28-01006-f001:**
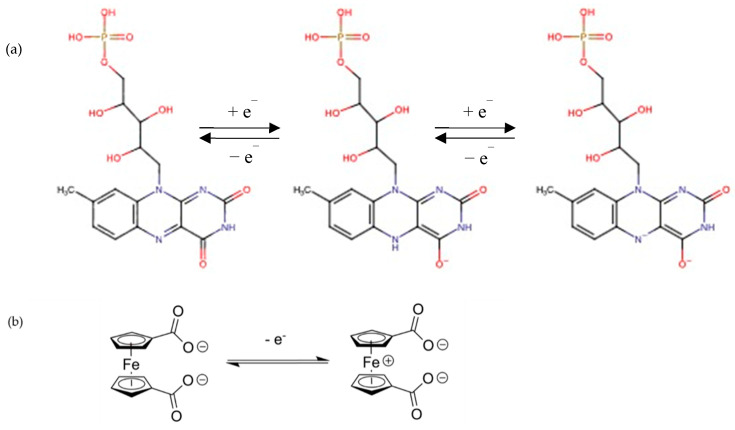
(**a**) Electrochemical transformation of riboflavin−5′−phospahate at pH neutral in aqueous media; (**b**) ferrocene dicarboxylic.

**Figure 2 molecules-28-01006-f002:**
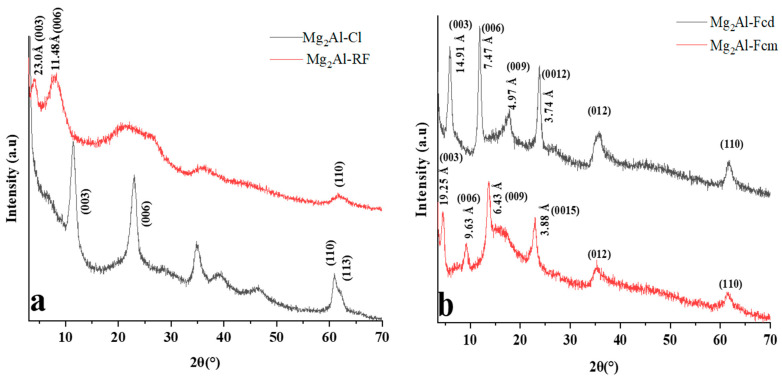
XRD of (**a**) LDH−RF and (**b**) LDH−FCm and LDH−FCm.

**Figure 3 molecules-28-01006-f003:**
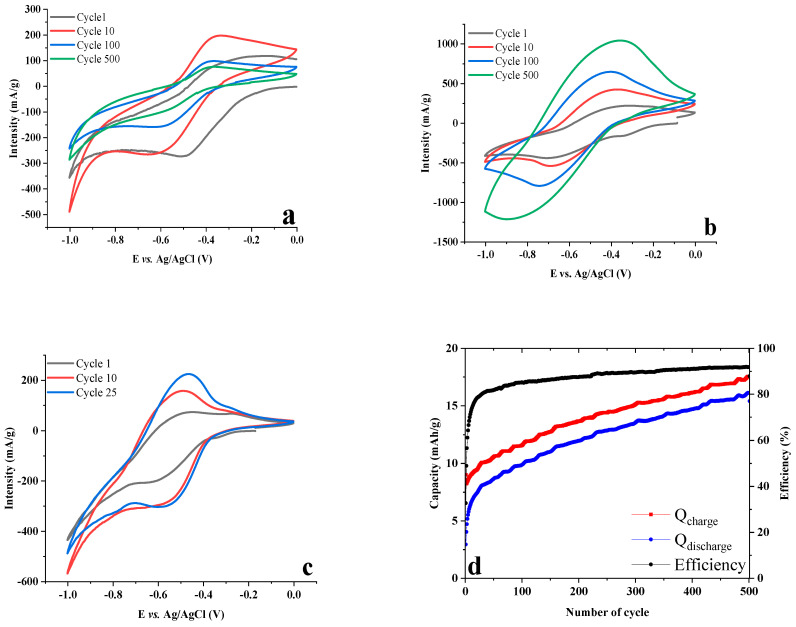
CV experiments of (**a**) RF, (**b**) LDH−RF at 10 mV/s in 1 M sodium acetate, (**c**) LDH−RF at 1 mV/s in 1 M sodium acetate and (**d**) capacity and efficiency of LDH−RF@10 mV/s, depending on the number of cycles.

**Figure 4 molecules-28-01006-f004:**
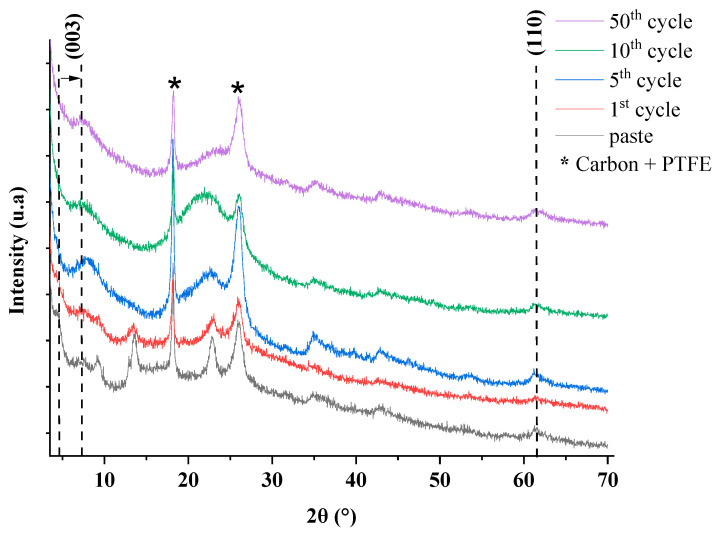
XRD of LDH−FCm electrode after 1st, 5th, 10th and 50th cycles in sodium acetate at 10 mV/s.

**Figure 5 molecules-28-01006-f005:**
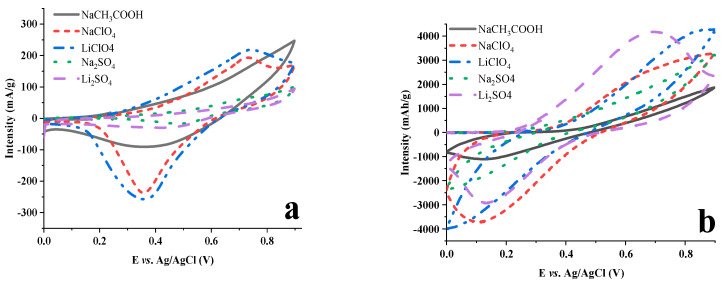
First CV experiments of (**a**) LDH−FCd and (**b**) LDH−FCm at 10 mV/s in different electrolyte.

**Figure 6 molecules-28-01006-f006:**
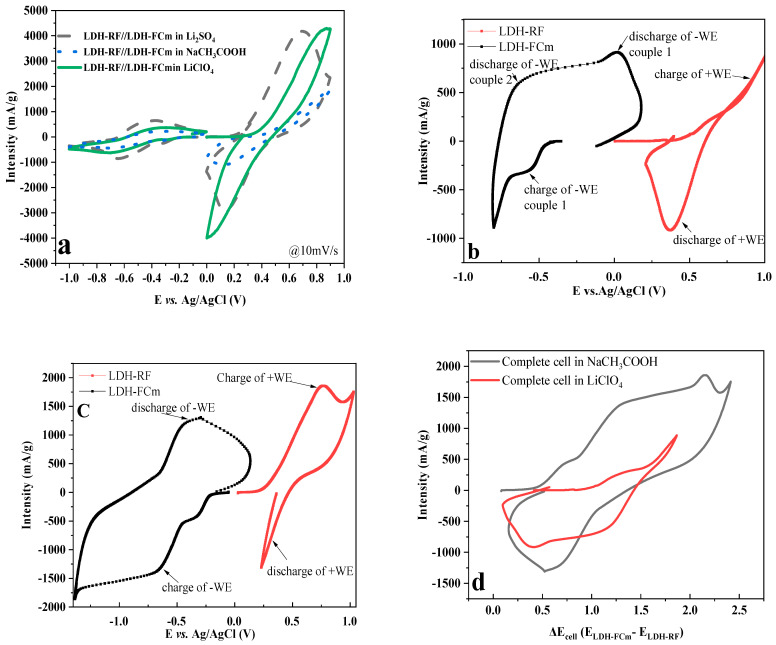
CV of (**a**) LDH−RF and LDH−FCm in 1 M sodium acetate sulfonate, lithium sulfate and lithium perchlorate at 10 mV/s; (**b**) complete cell d of LDH−FCm//LDH−RF in 1 M lithium perchlorate at 10 mV/s, (**c**) complete cell of LDH−FCm//LDH−RF in 1 M sodium acetate at 10 mV/s, (**d**) comparison of E_cell_ depending on the aqueous electrolyte salt, (**e**) capacity depending on cycle number for LDH−FCm//LDH−RF in 1 M sodium acetate at 10 mV/s, (**f**) capacity depending on cycle number for LDH−FCm//LDH−RF in 1 M lithium perchlorate at 10 mV/s (**g**) 1st cycle charge capacity as a function of the scan rate for the complete cell LDH−FCm//LDH−RF, the electrolyte salt is indicated.

**Table 1 molecules-28-01006-t001:** CHNS composition of LDH-RF, LDH-FCm and LDH-FCd.

Sample	%N_theo_	%N_exp_	ΔN(%)	%C_theo_	%C_exp_	ΔC(%)	%H_theo_	%H_exp_	ΔH(%)
LDH-RF	4.52	2.73	1.74%	30.52	19.07	0.46%	4.52	4.37	0.63%
LDH-FCm	/	/	/	20.60	20.44	0.15%	4.32	4.13	2.66%
LDH-FCd	/	/	/	29.84	29.48	0.27%	4.02	4.18	1.92%

**Table 2 molecules-28-01006-t002:** Results of CV experiments of LDH−RF in different electrolytes at 10mV/s.

Electrolyte	E_ox_ (V)	E_red_ (V)	Qc_1st cycle_ (mAh/g)/ Qd_1st cycle_ (mAh/g) *	C. Efficiency * (%)	Qc_50th cycle_ (mAh/g)/ Qd_50th cycle_ (mAh/g) *	C. Efficiency * (%)	Qc_500th cycle_ (mAh/g)/ Qd_500th cycle_ (mAh/g) *	C. Efficiency * (%)
NaCH_3_COOH	−0.37	−0.69	2.9/9.0	32	7.3/10.3	71	16/18	89
NaClO_4_	−0.41	−0.68	4.3/9.5	45	11.5/12.9	89	10.6/12.3	88
LiClO_4_	−0.33	−0.69	4.9/11.7	42	11.9/17.5	68	14.8/23.7	62
NaNO_3_	−0.44	−0.76	6.8/31.4	22	9.1/23.3	39	9.7/23.2	42
Na_2_SO_4_	−0.46	−0.69	5.8/13.2	44	6.2/17.2	36	6.6/20.3	33
Li_2_SO_4_	−0.41	−0.65	7.6/12.3	62	9.3/10.1	92	11.1/12.0	93

* Capacity and Columbic efficiency include all the possible parasitic reactions taking place during the cyclic voltammetry tests.

**Table 3 molecules-28-01006-t003:** Results of CV experiments of LDH−FCm and LDH−FCd in different electrolytes at 10 mV/s.

	Electrolyte	E_ox_ (V)	E_red_ (V)	Qc_1st_ Cycle	Qd_1st_ Cycle	Efficiency (%)	Qc_100th_ Cycle	Qd_100th_ Cycle	Efficiency (%)
LDH-FCd	NaCH_3_COOH	0.58	0.33	2.7	0.9	33	0.8	0.6	75
	NaClO_4_	0.67	0.36	2.3	1.3	57	1.1	1.0	91
	LiClO_4_	0.61	0.36	2.9	1.9	66	1.0	1.0	100
	Na_2_SO_4_	0.63	0.48	1.1	0.1	9	0.1	0.1	100
	Li_2_SO_4_	0.67	0.36	1.0	0.4	40	0.6	0.5	83
LDH-FCm	NaCH_3_COOH	0.57	0.25	19.1	14.7	60	3.2	2.3	71
	NaClO_4_	0.72	0.11	44.1	26.3	60	11.2	10.6	94
	LiClO_4_	0.81	0.12	57.2	37.3	65	6.5	5.9	91
	Na_2_SO_4_	0.74	0.10	41.0	11.0	27	1.1	0.1	9
	Li_2_SO_4_	0.70	0.13	40.5	22.1	41	0.7	0.4	57

## Data Availability

Not applicable.

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
