# Peer review of "Interleaved Electroactive Molecules into LDH Working on Both Electrodes of an Aqueous Battery-Type Device"

_molecules, 2023, doi:10.3390/molecules28031006_

Round 1

Reviewer 1 Report

This manuscript reported Interleaved Electroactive Molecules into LDH Working on Both Electrodes of An Aqueous High Power Battery. After careful evaluation of the manuscript, I recommend major revisions prior to publication in the Molecules Journal.

1.     The experimental methodologies and material preparation are not new. Hence, the novelty of the work should be discussed in the introduction section of the revised manuscript.

2.     The authors are advised to change the title from “Electroactive Molecules into LDH Working on Both Electrodes of An Aqueous High Power Battery” to some relevant titles because this work does not include any High Power Battery performance metrics.

3.     It is difficult to classify the morphology of the individual electrodes and the composites. The authors need to include the various magnified FE-SEM images of as-prepared materials with EDS mapping analysis in the revised version of the manuscript.

4.     The author should discuss the EIS spectrum with equivalent circuit fitting for as-prepared materials and discuss it in detail.

Author Response

1.The experimental methodologies and material preparation are not new. Hence, the novelty of the work should be discussed in the introduction section of the revised manuscript.

Indeed, the methodologies used here are not new but the concept of an electrochemical device composed of two LDH electrodes intercalated with electroactive molecules is innovative. We have therefore placed this idea in relation to the current literature in the introduction.

2.The authors are advised to change the title from “Electroactive Molecules into LDH Working on Both Electrodes of An Aqueous High Power Battery” to some relevant titles because this work does not include any High Power Battery performance metrics.

The performance in terms of capacity is not important enough to be considered as "High Power battery" so we changed the title to:  Interleaved Electroactive Molecules into LDH Working on Both Electrodes of an Aqueous Battery-type device.

3.It is difficult to classify the morphology of the individual electrodes and the composites. The authors need to include the various magnified FE-SEM images of as-prepared materials with EDS mapping analysis in the revised version of the manuscript.

SEM images have been realized after battery test (Figure S9 and S10) and show the stability of the composite paste. In addition, EDX mapping analyses have been performed and it allows to underline that the Fcm molecules (followed by Fe element analysis) are coming out the LDH structure (Figure S11), since there are no correlation between Mg, Al and Fe elements position (color code) anymore, while there is such correlation before electrochemical measurements.

4.The author should discuss the EIS spectrum with equivalent circuit fitting for as-prepared materials and discuss it in detail.

We did not perform EIS tests as we believe that these tests would not be significant on the composite paste but rather on the intercalated LDH material itself.

Reviewer 2 Report

Reviewer’s Comments:

The manuscript “Interleaved Electroactive Molecules into LDH Working on Both Electrodes of An Aqueous High Power Battery” is very interesting work. In this work, by selecting two electroactive species immobilized in a layered double hydroxide backbone (LDH) host, one able to act as a positive electrode material and the other as a negative one, it was possible to match their capacity to design an innovative energy storage device. Each electrode material is based on electroactive species, riboflavin phosphate on one side and ferrocene carboxylate (FCm) on the other, both interleaved into a layered double hydroxide (LDH) host structure to avoid any possible molecule migration and instability. The intercalation of the electroactive guest molecules is demonstrated by X-ray diffraction with the observation of an interlayer LDH spacing of about 2 nm in each case. When successfully hosted into LDH interlayer space, the electrochemical behavior of each hybrid assembly was scrutinized separately in aqueous electrolyte to characterize the redox reaction occurring upon cycling. However, the following issues should be carefully treated before publication.

1. In abstract, the author should add more scientific findings.

2. Keywords: the synthesized system is missing in the keywords. So, modify the keywords.

3. In the introduction part, the introduction part is not well organized and cited references should cite recently published articles such as  10.1039/C8DT03107J, 10.1007/s11664-019-06929-w

4. Introduction part is not impressive and systematic. In the introduction part, the authors should elaborate the scientific issues in the battery research.

5. Electrochemical behavior of LDH-RF…, The author should provide reason about this statement “The reversible redox waves separate upon cycling, DE1 = 0.24 vs. 132 DE100 = 0.30 V, underlining that the redox process is increasingly more difficult and associated with a large decrease in capacity”.

6. The authors should explain regarding the recent literature why “The reversibility is better and the redox reduction wave is still centered at -0.5 V vs. Ag/AgCl after 25 cycle (Figure 3c)”.

7. Material and method: Materials “write all the detail of chemicals in unique format rather than to write individual chemical such as Tin Chloride (SnCl4.5H2O). It should be written as “tin chloride (SnCl4.5H2O, 98%, Sigma)”. Write all the chemicals in this format.

8. In order check the stability of the synthesized materials, the authors should provide the TEM images after battery test.

9. Comparison of the present results with other similar findings in the literature should be discussed in more detail. This is necessary in order to place this work together with other work in the field and to give more credibility to the present results.

10. The conclusion part is very week. Improve by adding the results of your studies.

Author Response

1.In abstract, the author should add more scientific findings.

We add the mode of redox reaction for both electrode materials. However, we did not provide additional values in terms of performance of capacity as they are dependent of the electrolyte salt.

  1. Keywords: the synthesized system is missing in the keywords. So, modify the keywords.

We add the synthesized system

3.In the introduction part, the introduction part is not well organized and cited references should cite recently published articles such as  10.1039/C8DT03107J, :  Electrochemical study of specially designed graphene-Fe3O4-polyaniline nanocomposite as a high-performance anode for lithium-ion battery† ; Ali Bahadur,‡*a   Shahid Iqbal,‡b   Muhammad Shoaib    a  and  Aamer Saeed*a 10.1007/s11664-019-06929-w , : Hussain, W., Malik, H., Hussain, R.A. et al. Synthesis of MnS from Single- and Multi-Source Precursors for Photocatalytic and Battery Applications. J. Electron. Mater. 48, 2278–2288 (2019). https://doi.org/10.1007/s11664-019-06929-w.

Although these articles are interesting to show the diversity of materials used for energy storage we feel that they are too far from the subject of the article and therefore not really appropriate. Indeed, these articles deal with phases which are for applications in lithium-ion batteries. We have therefore added more references to HDL phases as electrodes for storage devices and also more generally for the recent researches on battery materials using aqueous electrolyte.

4.Introduction part is not impressive and systematic. In the introduction part, the authors should elaborate the scientific issues in the battery research. 

We have carried out more extensive bibliographical researches, particularly on the use of electroactive molecules directly as electrolyte salts, but also on the use of LDH phases in various energy storage devices as well as the problems encountered in these works.

  1. Electrochemical behavior of LDH-RF…, The author should provide reason about this statement “The reversible redox waves separate upon cycling, DE1 = 0.24 vs. 132 DE100 = 0.30 V, underlining that the redox process is increasingly more difficult and associated with a large decrease in capacity”.

The reversible redox waves separate upon cycling, DE1 = 0.24 vs. DE100 = 0.30 V, underlining that the redox process is increasingly more difficult, this is due to the fact that part of the riboflavin molecules are no longer involved in the reversible redox processes and associated with a large decrease in capacity. RF is released from the interlayer space and the electrode into the electrolyte, resulting in the loss of capacity.

  1. The authors should explain regarding the recent literature why “The reversibility is better and the redox reduction wave is still centered at -0.5 V vs. Ag/AgCl after 25 cycle (Figure 3c)”.

 As indicated in the figure caption, the scan rate is decreased for fig. 3c from 10 mV/s down to 1 mV/s, that may explain the reason why the redox reaction is more superimposed (either in reduction or in oxidation). This is the case after the 10 first cycles, however, a pronounced decrease in intensity is observed after the 25th cycle.

We add the references concerning the electrochemical behavior of riboflavin (see manuscript), but there is no behavior in cycling.

As there is no relation between “the reversibility is better” and “the location of the redox reduction wave”, we remove the first part of the sentence.

  1. Material and method: Materials “write all the detail of chemicals in unique format rather than to write individual chemical such as Tin Chloride (SnCl4.5H2O). It should be written as “tin chloride (SnCl4.5H2O, 98%, Sigma)”. Write all the chemicals in this format.

We change accordingly for all the chemicals.

  1. In order check the stability of the synthesized materials, the authors should provide the TEM images after battery test.

SEM images have been realized after battery test (Figure S9 and S10) and show the stability of the composite paste. In addition, EDX mapping have been performed too and underline the leaving of FCm out of the LDH structure (Figure S11).

  1. Comparison of the present results with other similar findings in the literature should be discussed in more detail. This is necessary in order to place this work together with other work in the field and to give more credibility to the present results.

We agree to compare the results obtained in a complete cell with the results of the literature. However, there is no article reporting such a system (two LDH electrodes intercalated with two electroactive molecules operating in an aqueous electrolyte). The systems studied in the literature only use an HDL-based electrode (type CoFe, NiCoFe, NiMnFe) without electroactive species intercalated. The capacities obtained are therefore much higher than that we obtain here from organic molecules. We think that it would be more judicious to show the possibility of realizing such a system and not the sole performances.

  1. The conclusion part is very week. Improve by adding the results of your studies.

We add the results of half-cell test and results with the lithium perchlorate

Reviewer 3 Report

In this work the authors prepare two composites containing different electroactive materials intercalated in layered double hydroxide hosts and study their electrochemical behavior. Although the intercalation of ferrocene carboxylate derivatives in LDHs is not new, the intercalation of the riboflavin derivative has not been previously reported. Finally the authors show that the two electrode materials can be combined to fabricate a prototype aqueous battery system. This work is appropriate for publication after the following issues are addressed:

(1)   Although the authors show the general formula for a riboflavin in Figure 1, they should also give the formula for the specific riboflavin-5-phosphate that they actually used.

(2)   The authors need to explain the relationship between the experimental and calculated CHN analytical data in Table 1 and the formulae they propose in the text following the table (lines 116-119). In particular (i) given that the loading of riboflavin-5-phosphate appears to be much lower than expected, could it be intercalated as a mixture of mono- and dications? (ii) Given that the C:N ratio in the product is much higher than in the riboflavin-5-phosphate (C17N4) could there be some co-intercalation of carbonate anions? (iii) the second %Ntheo is presumably a typo – should be %Htheo?

(3)   Any change to the proposed formula of LDH-RF will affect the estimation of its theoretical capacity (line 142) since this is based on 2e/formula weight.

(4)   Given that MgAl LDH hosts are used in this work, the authors need to justify their comment about the reversible oxidation/reduction of the cations in the.LDH layers (line 180): the reference cited in support of this statement (ref. 21) refers to a ferrocene-functionalized graphene oxide nanocomposite.

Author Response

  1. Although the authors show the general formula for a riboflavin in Figure 1, they should also give the formula for the specific riboflavin-5-phosphate that they actually used.

            We have replaced the Figure 1 with the riboflavin-5-phosphate.

  1. The authors need to explain the relationship between the experimental and calculated CHN analytical data in Table 1 and the formulae they propose in the text following the table (lines 116-119). In particular

(i) given that the loading of riboflavin-5-phosphate appears to be much lower than expected, could it be intercalated as a mixture of mono- and dications?

We add that it was difficult to intercalate molecules of this type (large size, aromatic, causing steric interaction problems). The co-precipitation method is most probably not the best way to intercalate these molecules. In addition, we have intercalation of other molecules as evidenced by TGA-MAS with the presence of carbonate anions which have a strong affinity with LDH host.

(ii) Given that the C:N ratio in the product is much higher than in the riboflavin-5-phosphate (C17N4) could there be some co-intercalation of carbonate anions?

It is our mistake. Indeed, using the ATG-MS data, there is indeed a peak at 44g/mol corresponding to a non-negligible release of CO2 (Fig. S1) linked to carbonate anions intercalated in the HDL matrix. This explains the difference in the C/N ratio with a higher presence of the carbon chemical element. For the calculation of the compositions we consider the amount in N chemical element to calculate the percentage of RF in the HDL structure, then the rest of the anions to compensate the charge are therefore carbonate anions (as evidenced by ATG-MAS) and not chloride anions as previously mentioned.

The correct formula is then Mg2Al(OH)6(CO32-)0.185(RF)0.63  . 3.1 H2O

(iii) the second %Ntheo is presumably a typo – should be %Htheo?

Indeed, it is a typing error

  1. Any change to the proposed formula of LDH-RF will affect the estimation of its theoretical capacity (line 142) since this is based on 2e/formula weight

Qtheo=71.6 mAh/g and we have change all the data correlated with the change above (response 2ii).

  1. Given that MgAl LDH hosts are used in this work, the authors need to justify their comment about the reversible oxidation/reduction of the cations in the.LDH layers (line 180): the reference cited in support of this statement (ref. 21) refers to a ferrocene-functionalized graphene oxide nanocomposite.

The layers composed by Mg2+ and Al3+ cations remain stable and not electrochemically active in the operating conditions (potential range and electrolyte). Consequently, the redox process generated in the interlayer space destabilizes the LDH-organic molecule assembly leading to their de-intercalation out of the host structure (as evidenced by SEM mapping) and most probably to the ingress of anions (coming from the electrolyte) into the host to equilibrate again the layer charge. 

Round 2

Reviewer 1 Report

I read through the revised version and the authors' responses. Overall, this revised version is notable. After careful evaluation of the manuscript, I recommend minor revisions prior to publication in the Molecules Journal.

1.     All abbreviations in the manuscript should be given full names when they first appear.

2.     Some recent progresses of supercapacitors shall be included in the introduction (ACS Appl. Mater. Interfaces 2021, 13, 15315–15323, Ionics 28, 859–869 (2022); Energy Fuels 2021, 35, 16, 13438–13448, J Mater Sci: Mater Electron 33, 8426–8434 (2022); Journal of Alloys and Compounds, 882, 15, 2021, 160409.

3.     The authors should study elemental analysis it will confirm the purity of the sample.

Author Response

  1. All abbreviations in the manuscript should be given full names when they first appear.

We have made all the necessary changes to meet this request. In particular, the following acronyms as RF, AQS, FCd, FCm, etc… are now defined in the text.

  1. Some recent progresses of supercapacitors shall be included in the introduction (ACS Appl. Mater. Interfaces 2021, 13, 15315–15323, Ionics 28, 859–869 (2022); Energy Fuels 2021, 35, 16, 13438–13448, J Mater Sci: Mater Electron 33, 8426–8434 (2022); Journal of Alloys and Compounds, 882, 15, 2021, 160409.

Although these articles are interesting to show the diversity of materials used for energy storage we feel that some of them are too far from the subject of our article but we select two articles of interest: Ionics 28, 859–869 (2022) from Rajkumar et al., is dealing with aminophenol-modified ZnO in acid electrolyte, and Journal of Alloys and Compounds, 882, 15, 2021, 160409 about CeO2 doped Ni particles, showing high cyclic stability but with rare-earth element.

  1. The authors should study elemental analysis it will confirm the purity of the sample.

We have already performed the CHNS elemental analysis and determined that the LDH-RF phase was not pure (co-intercalation of carbonate proved by a C/N ratio higher than that of pure RF and a peak at 44g/mol on the TGA/MS data). For the LDH-FC phases, we can confirm that the phase is totally intercalated by the electroactive molecule. Moreover, the CV data at 1mV/s can support this as 100 % of the theoretical capacity is obtained (at the 1st cycle for LDH-FCm and at the 25th cycle for LDH-RF taking into account the carbonate anions co-intercalated for the latter composition).